# ScaleNAS: Multi-Path One-Shot NAS for Scale-Aware High-Resolution Representation

**Hsin-Pai Cheng**[1] **Feng Liang**[1] **Meng Li**[2] **Bowen Cheng**[3] **Feng Yan**[4] **Hai Li**[1] **Vikas Chandra**[2] **Yiran Chen**[1]

[1]Duke University
[2]Meta Reality Labs
[3]University of Illinois Urbana-Champaign
[4]University of Nevada, Reno

**Abstract** Scale variance among different sizes of body parts and objects is a challenging problem for visual recognition tasks. Existing works usually design a dedicated backbone or apply Neural architecture Search (NAS) for each task to tackle this challenge. However, existing works impose significant limitations on the design or search space. To solve these problems, we present ScaleNAS, a one-shot learning method for exploring scale-aware representations. ScaleNAS solves the limitation of scale representation by searching multi-scale feature aggregation. ScaleNAS adopts a flexible multi-path search space that allows an arbitrary number of blocks and cross-scale feature fusions. To cope with the high search cost incurred by the flexible space, ScaleNAS employs one-shot learning for multi-scale (multi-path) supernet driven by grouped sampling and evolutionary search. Without further retraining, ScaleNet can be directly deployed for different visual recognition tasks with superior performance. We use ScaleNAS to create high-resolution models for two different tasks, ScaleNet-P for human pose estimation and ScaleNet-S for semantic segmentation. ScaleNet-P and ScaleNet-S outperform existing manually crafted and NAS-based methods in both tasks. Using ScaleNet-P for bottom-up human pose estimation, it achieves a new state-of-the-art on COCO test-dev and CrowdPose test. In particular, ScaleNet-P4 achieves 71.3% AP on CrowdPose test, surpassing the previous best result by a large 3.7% AP margin.

## 1 Introduction

Deep representation learning can be generally categorized into low-resolution representation learning and high-resolution representation learning. Low-resolution representation is typically used in classification tasks while high-resolution representation is essential for visual recognition tasks such as semantic segmentation and human pose estimation. We focus on the high-resolution representation in this paper. There are three important yet challenging considerations when designing high-resolution representation: 1) the scale variance from different sizes of objects and scenes is quite large; 2) precise and informative feature maps are critical; 3) the expensive cost in training and searching multi-path architectures.

1) **Challenge of scale variance**: Take semantic segmentation as an example, the variance of object size induces difficulty for pixel-level dense prediction, and thus scale-aware representation is critical. In human pose estimation, it is challenging to localize human anatomical keypoints when there is a high scale variance in the scene such as people having different sizes, the large difference in joint distance. We address this challenge by proposing a new multi-scale search space.

2) **Challenge of high-resolution representation:** To design high-resolution representations, earlier efforts recover high-resolution representations from low-resolution outputs, e.g., Hourglass [27], SegNet [3], U-Net [32]. Recent works focus on maintaining high-resolution representation through the whole network and aggregating different scales of representation from parallel

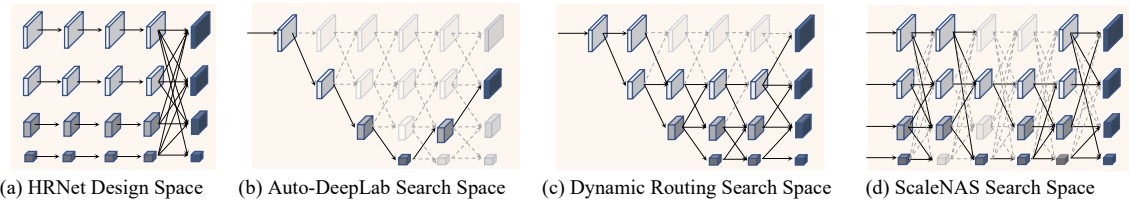

(a) HRNet Design Space  (b) Auto-DeepLab Search Space  (c) Dynamic Routing Search Space  (d) ScaleNAS Search Space

Figure 1: Search space comparison. (a) HRNet uses fully connected multi-scale feature fusion. (b,c) Auto-DeepLab and dynamic routing allow neighbor feature fusion connection. (d) We propose a flexible feature fusion that allows crossing to remote feature maps to maximize multi-scale aggregation.

paths, e.g., HRNet [33, 34] and its variants. However, multi-scale neural architectures usually have a large design space to explore and are prone to design redundancies. Our study reveals that when different scales of representation have different depths, the performance can be greatly improved. We push the envelope even further by exploring scale-aware representations in a much more flexible design space.

3) **Challenge of the expensive cost in training and searching multi-path architectures:** In order to derive a wide spectrum of models, training and searching cost is prohibitively expensive. Among various searching methods, one-shot based NAS adopts weight-sharing mechanism to reduce the searching cost [24, 16]. State-of-the-art *one-shot searching without retraining* allows us to search multiple well-performed neural architectures at a considerably lower searching cost [6, 38]. However, current one-shot based searching[1] [6, 38] is only limited to single-path [6, 38] due to the difficulty and high cost in supernet training [12]. Considering the high-resolution scale-aware representation with multi-scale aggregation is very useful for visual recognition tasks, we are motivated to enhance the one-shot based searching method with multi-path capability. Since training a multi-path architecture intrinsically has more weights to be maintained, the existing multi-path one-shot method [12] is only limited to DARTS [24] search space without multi-scale setting in each stage.

To address the above challenges, we propose ScaleNAS, a multi-path one-shot based searching method to explore scale-aware neural architectures. We tackle the scale variance challenge by proposing *multi-scale aggregation search space* to explore multi-scale aggregation and network depth for high-resolution representation. While this new search space enables a more flexible neural architecture design with better information integration, it also brings new challenges for the search method. Specifically, such a significantly large search space causes sub-optimal accuracy of sub-networks due to inefficient network exploration. To enable efficient searching for the proposed multi-path search space, we further establish a new searching method to discover multiple architectures simultaneously. Specifically, we introduce two techniques grouped sampling and multi-scale topology evolution to help the supernet training and architecture searching, respectively.

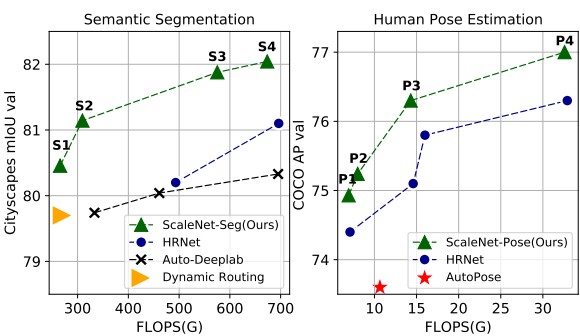

Figure 2: The trade-off between computation cost (GFLOPs) and model performance. Left: semantic segmentation mIoU on Cityscapes val. Right: human pose estimation AP on COCO val. Our ScaleNet outperforms HRNet and NAS-based methods.

We name these elite architectures ScaleNet. As demonstrated in Figure 2, ScaleNet outperforms manually crafted and NAS-based models on semantic segmentation and human pose estimation. In semantic segmentation, ScaleNet surpasses HRNet, Auto-DeepLab, and dynamic routing by 1.3% -

---

[1] In this paper, one-shot NAS refers in particular to NAS without retraining.

1.7% mIoU on CityScapes dataset with less computation cost. In bottom-up human pose estimation, our ScaleNet-P4 achieves a new state-of-the-art on COCO *test-dev2017* and CrowdPose *test*. In particular, ScaleNet-P4 achieves 71.3% AP on CrowdPose *test*, surpassing the previous best result by a large 3.7% AP margin.

## 2 Related Work

**NAS for High-resolution Architectures.** Recent works proposed to automate the design of neural architectures for semantic segmentation and human pose estimation. For example, Auto-DeepLab and dynamic routing [21] are proposed to search architectures for semantic segmentation models. PoseNAS [4], AutoPose [15] and PNFS [36] are proposed to search architectures for human pose estimation. However, the above NAS-based methods can only craft architectures for one task at a time. Our ScaleNAS supports dealing with multiple tasks simultaneously by searching multi-scale feature aggregation in a novel multi-path search space. In addition, previous works [14, 21] are based on a limited multi-scale fusion constraint that each of the scale can only connect to its neighboring scale. ScaleNAS adopts a more flexible cross-scale fusion to allow better information gathering resulting in higher accuracy. Moreover, existing methods can only derive one architecture at a time which leads to $O(N)$ time to derive $N$ models. We propose multi-path one-shot based searching method to lower the total cost to $O(1)$.

**Multi-Path One-Shot Neural Architecture Search.** Weight-sharing NAS [29, 5, 24, 7, 16] aims at deriving neural architectures from an over-parameterized neural network (supernet). One-shot NAS can be generally categorized to single-path one-shot (SPOS) and multi-path one-shot (MPOS) [12].

For SPOS, recent works focus on deriving a family of state-of-the-art neural architectures without extra retraining or post-processing [6, 38]. For example, BigNAS [38] transforms the problem of training supernet to training a big single-stage model and applies sandwich rule to guarantee the performance for each sub-network. OFA [6] proposes to use progressive shrinkage together with distillation to train a one-shot supernet. However, these methods are mainly designed for single-path neural architecture on relatively simple task (e.g., ImageNet classification).

For MPOS, it intrinsically has more trainable parameters to be maintained. As each of the sub-network share its weights on supernet, the weight interference from sub-networks and training instability are significantly amplified in the multi-path setting [12]. Thus, directly applying MPOS to multi-path search space and more complicated tasks (e.g., segmentation and pose estimation) is not a feasible solution. In addition, the exponential increase of search space size makes multi-scale supernet training and architecture searching even more challenging.

In this paper, we take the advantage of the diversity of multi-path search space and develop new supernet sampling and evolutionary approaches to improve the searching method to enable exploring a wide spectrum of sub-networks as well as fast searching of elite architectures.

## 3 Search Space Design

One important observation is that the search spaces in existing work (Figure. 1) have multiple redundancies and lack of representation ability. To address this search space problem, we first perform a search space exploration. Then we introduce a new *multi-scale aggregation search space* to overcome the limitations of exiting search space.

### 3.1 Search Space Exploration

Existing works that achieve state-of-the-art results on semantic segmentation and human pose estimation impose limitations on the design space. As shown in Figure 1, HRNet supports cross-scale feature fusions, but it uses four residual blocks in every scale of branchs. Such regular design results in redundancy and misses optimization opportunities as the depth for each branch can be altered to improve performance.

Auto-DeepLab [23] includes multiple scale options in their search space to search a single-path neural architecture for semantic segmentation. Dynamic routing [21] reused the search space form Auto-DeepLab to search a multi-path neural architecture to achieve improved performance on semantic segmentation. Although Auto-DeepLab and dynamic routing search the connections between different scales of feature maps, the search space for fusion is limited to only neighboring scales. Such limitation restricts the representation ability for feature maps and cross-scale feature fusion can provide better information gathering for each scale of feature maps. To illustrate our search space, we compare ScaleNet search space with HRNet and dynamic routing search space. Different from existing works, we provide flexible depth (number of residual blocks) for each scale of branch. In addition, we allow feature fusion to cross to any other scale of branches. We randomly sample several architectures from our ScaleNet search space and dynamic routing search space, and train them on Cityscapes [13]. With the same random sample size, ScaleNet search space achieves higher average accuracy than dynamic routing search space and HRNet (Figure 3). This reveals the huge potential of multi-scale feature aggregation in our ScaleNet search space.

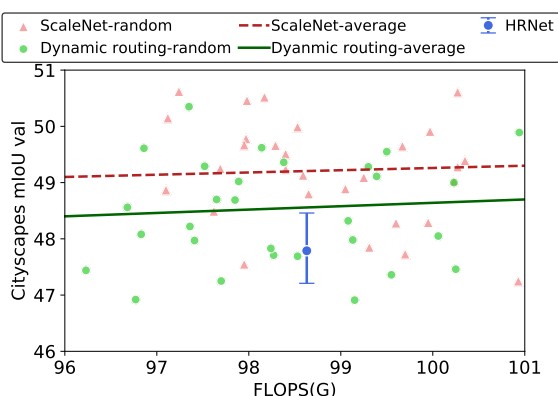

Figure 3: Search space exploration. ScaleNet-random and Dynamic routing-random denotes randomly selected architectures. ScaleNet-average and Dynamic routing-average denotes the average accuracy of each group.

## 3.2 Multi-Scale Aggregation Search Space

Based on the experimental results shown in Figure 3, we observe that cross scale feature fusion provides more network diversity and better feature integration that comes with higher accuracy. Thus we propose a new multi-scale aggregation search space and shows its overview in Figure 4.

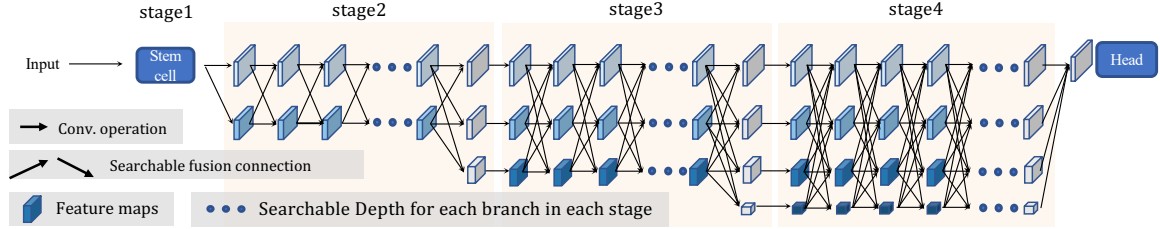

Figure 4: ScaleNAS search space overview. Our search space inherit the spirit of HRNet that has few stages. ScaleNAS adopts a flexible search space with arbitrary number of blocks and cross-scale feature fusions.

We employ a stage-based search space design, which is inspired by the state-of-the-art architecture HRNet that can be adapted to multiple tasks. Our search space starts from a stem cell as the first stage and it is composed of two stride-2 3×3 convolutions. There are four stages in our search space. After the first stage, we gradually add one more high-to-low branch for the following stages, i.e., stage2, stage3, and stage4 maintains two-resolution, three-resolution, and four-resolution branches respectively. As shown in Figure 4, the convolution operation in our search space is the residual block which is composed of two $3 \times 3$ convolutions. Searchable fusion includes downsampling and upsampling. For downsampling, we use strided $3 \times 3$ convolution with stride 2. For upsampling, we use bilinear upsampling followed by a $1 \times 1$ convolution for aligning the number of channels [34].

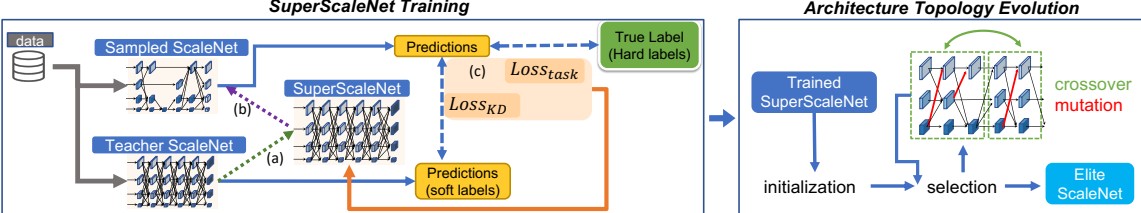

Figure 5: Workflow of ScaleNAS. ScaleNAS train a SuperScaleNet in the proposed search space and uses our proposed evolutionary method to explore elite architectures based on the trained SuperScaleNet. (a) Before training starts, we initialize SuperScaleNet by the teacher model. (b) During each iteration, we sample ScaleNet from the SuperScaleNet. (c) We use the task loss from true labels and the distillation loss from soft labels given by teacher to update SuperScaleNet.

We introduce two controlling factors to form our search space:

1. Branch depth ($d$). Instead of searching the overall depth for the entire network, we allow a more flexible search space that can search depth for each branch in individual independent stage module. For simplicity, the depth of each branch is chosen from {d1, d2, d3, d4}. In this paper, we currently set it at {2, 3, 4, 5}

2. Fusion percentage ($f$). Here the fusion percentage is defined as the probability of the out-degree fusion for each feature map. E.g., a feature map with fusion percentage of 50% means this feature map randomly connects to half of the other scales of feature map at the next depth.

By relaxing the cross-scale feature fusion and enlarging the branch blocks, we have roughly $7 \times 10^{72}$ different neural network architectures in our search space.

## 4 Architecture Search

Our goal is to design multi-scale neural architectures that can be adapted to various visual recognition tasks without retraining. Instead of searching a single model at a time, we aim at discovering a wide spectrum of models that have different computation cost for different deployment scenarios. Meanwhile, we also need to address the inefficiency of network exploration brought by the new multi-path search space. To explain how our multi-path one shot searching method achieve the above goals, we first introduce how to train a *one-shot based SuperScaleNet* that contains a wide spectrum of architecture candidates. Then, we employ *multi-scale topology evolution* to derive elite ScaleNet based on a trained SuperScaleNet.

### 4.1 Training One-Shot SuperScaleNet

The configuration of SuperScaleNet follow the description in Figure 4 with maximum number of depth ($d = 5$) and fully connected feature fusion ($f = 100\%$). Figure 5 depicts the workflow of SuperScaleNet training. We follow the common practice of SPOS [6, 38, 11] to use the largest architecture as a teacher model to provide soft labels during each iteration. In our case, the teacher model is the same configuration as SuperScaleNet.

Visual recognition tasks usually rely on ImageNet pretrained to stabilize training [34, 31], we pretrain the teacher model on ImageNet until converge. Then, we initialize SuperScaleNet with the weights from the teacher model. During each training iteration, we sample a sub-network from SuperScaleNet, pass one batch of training data to both sampled sub-network and teacher model. Next, we calculate the task loss, using the true label, distillation loss using the soft label given by teacher model. Finally, we update the supernet based on the combination of both task loss and

distillation loss. Our training objective is formulated as follow:

$$\min_{W_s} \sum_{arch_i} (\mathcal{L}_{task}(P_{arch_i}, y) + \alpha \cdot MSE(P_{arch_i}, P_t)). \tag{1}$$

Our main goal is to optimize the weights of SuperScaleNet $W_s$ with the combination of true label loss and soft label loss. Here $P_{arch_i}$ stands for the prediction for each sampled architecture, $y$ is the true label. $P_t$ stands for the prediction from teacher model. We use mean squared error (MSE) to calculate the loss between sub-network prediction and teacher network prediction. The distillation ratio $\alpha$ is set to 1.

**Grouped Sampling.** To train supernet, sampling plays a crucial role. Sandwich rule [37, 38] is a common way to train supernet, where the smallest model, the largest model, and two randomly sampled models are trained in every iteration. However, we observe that sandwich rule cannot guarantee to explore a wide spectrum of neural architectures especially in multi-path search space. Therefore, we propose grouped sampling by dividing the whole search space into different sub-groups. According to central limit theorem (CLT) [2], the distribution of sampled architectures $D$ can be approximated by a normal distribution with a mean and variance, noted as $\mu$ and $\sigma$. Directly applying the SPOS based sampling method (sandwich rule), the sampled architecture is mostly centered at the mean of $\mu_0$. Therefore, previous methods cannot fully explore the wide search space as denoted in Figure 6.

However, our grouped sampling approach overcomes such limitation by dividing the large search space into several sub-search-spaces. Grouping the entire search space to $i$ sub-groups, we essentially allow the supernet to explore each of the sub-search-space with different distribution characterized by different mean values, e.g., $\mu_1, \mu_2, \ldots, \mu_i$.

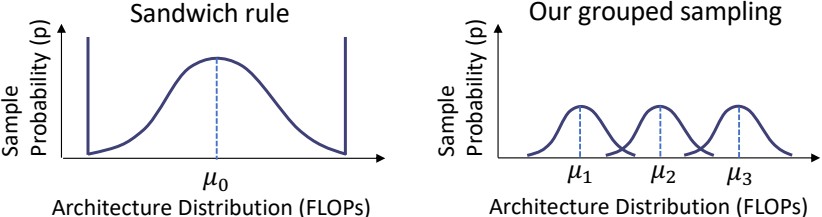

Figure 6: The probability density function of architecture sampling. Sandwich rule samples the smallest, the largest, and two networks from the search space with distribution mean of $\mu_0$. Our grouped sampling divides the large multi-scale search space to sub-groups and perform sampling from each distribution.

Specifically, given a depth choice of {d1, d2, d3, d4}, we group the depth choice to {[d1, d2], [d2, d3], [d3, d4]} (e.g., {[2, 3], [3, 4], [4, 5]}), the fusion percentage is selected from {f1, f2, f3} (e.g., {0.2, 0.5, 0.8}). In combination, we have a total of 9 (3×3) sub-groups. Compared with sandwich rule, grouped sampling is more suitable for multi-scale aggregation search space. Empirical justification is detailed in Section 5.3.

It is worth noting that unlike in OFA [6] or BigNAS [38], where 4 sub-networks are sampled and their gradients are aggregated in each update step, our group sampling only get one sub-network in each iteration. Therefore, the cost of training a SuperScaleNet using group sampling is very low, e.g., equivalent to a standard model training.

## 4.2 Multi-Scale Architecture Topology Evolution

To explore sub-network from a trained supernet, existing one-shot based methods use coarse-to-fine selection [38], evolutionary method [6], etc. However, these methods are mainly designed for single-path neural architecture. In comparison, our search space includes multiple path in each stage and each stage has different number of scales. Therefore, we propose *multi-scale architecture*

*topology evolution*, which provides more reasonable and controllable crossover and mutation over candidate architectures. As described in Figure 5, our topology evolution include the following four phases. The detailed algorithm can be found in the supplementary materials.

- **Step1 Initialization.** We uniformly sample $n_0$ sub-networks and record their architectures, accuracies, and FLOPs as a set **D**. $n_0$ equals to 1,000 in our experiments.

- **Step2 Selection.** We select top $k$ models on the Pareto front of cost/accuracy trade-off curve in **D** as candidate group **C**. For each subnetwork in **C**, we do crossover and mutation to obtain next-generation offsprings. $k$ is set to 100 in our experiments.

- **Step3 Crossover.** Since different stage module may have different number of branches, our crossover is inner-stage crossover. For each sub-network $\mathbf{arch_c}$ in **C**, we allow a probability $p_c$ to swap stage module settings with another randomly selected sub-network. There are 8 stage modules (1,4,3 for stage 2, 3, 4, respectively) and $p_c$ is set to 0.25. Thus, each sub-network is expected to have 2 modules been replaced.

- **Step4 Mutation.** After crossover, we do random mutation to switch on and off the fusion connections in every stage module with probability $p_m$. $p_m$ is set to 0.5. These $K$ offspring models along with their corresponding accuracies and FLOPs are recorded as set **M**. Then we update **D** = **D** ∪ **M**. We continue to Step2 until we have $N$ sub-networks in **D**. Here we set $N$ as 2,000.

The advantage of our method is twofold: 1) our evolution method maintains a large number of Pareto front models (100) as parent groups while aging evolution only has one parent in every generation which limits the search space exploration; 2) our cross-over and mutation allow us to explore macro and micro architectures in the multi-scale search space. Specifically, our cross-over serves as a macro architecture search that explores the order of each inner stage modules. Our mutation provides a micro architecture search by tweaking feature fusion connections. Empirical justification is detailed in Section 5.3.

## 5 Experiments

In this section, we evaluate ScaleNAS by searching neural architectures for semantic segmentation and human pose estimation. First we train SuperScaleNet on semantic segmentation with Cityscapes dataset [13] and derive ScaleNet-S. Then we apply the same searching routine on top-down human pose estimation framework with COCO dataset [22] to derive ScaleNet-P. In order to evaluate the generalizability of ScaleNet, we apply ScaleNet-P to HigherHRNet framework for bottom-up human pose estimation. Finally, we conduct analysis and ablation studies.

### 5.1 Semantic Segmentation

We conduct semantic segmentation tasks on Cityscapes [13] dataset. Table 1 reports the comparison between ScaleNAS and existing manual/NAS methods on semantic segmentation. Comparing with NAS (Auto-Deeplab and dynamic routing), ScaleNAS is much more efficient for multiple deployment scenarios. E.g., when there are 40 deployment scenarios, the total cost of ScaleNAS is 12× fewer than dynamic routing and 19× fewer than Auto-Deeplab, respectively. Without additional retraining, ScaleNet-S1 outperforms the dynamic routing Layer33-PSP by a 0.8% margin under the similar cost. When comparing with manually designed HRNet-W48 or Searched-F48-ASPP, ScaleNet-S4 improves the mIoU to 82.0%, surpassing HRNet and Auto-Deeplab by 0.9% and 1.7% respectively.

### 5.2 Human Pose Estimation

For human pose estimation, we first search ScaleNet-P on top-down human pose estimation task using COCO [22]. Then we reuse ScaleNet-P on MPII [1] and bottom-up pose estimation tasks.

Table 1: Semantic segmentation results on Cityscapes *val* (single scale, no flipping). GFLOPs is calculated on the input size $1024 \times 2048$. 'D-X' equals to 'Dilated-X'. For existing segmentation NAS works, the cost grows linear to the number of deployment scenarios $N$, while ScaleNAS cost remains constant.

| Method | Backbone | #Params | GFLOPs | mIoU (%) | Searching Cost (GPU hours) | Training Cost (GPU hours) | Total Cost($N$=40) (GPU hours) |
|---|---|---|---|---|---|---|---|
| DeepLabv3 [8] | D-ResNet-101 | 58.0M | 1778.73 | 78.5 | - | 50$N$ | - |
| DeepLabv3+ [9] | D-Xception-71 | 43.5M | 1444.63 | 79.6 | - | - | - |
| PSPNet [39] | D-ResNet-101 | 65.9M | 2017.63 | 79.7 | - | 100$N$ | - |
| Auto-DeepLab [23] | Searched-F20-ASPP | - | 333.3 | 79.7 | 72$N$ | 250$N$ | 12.9$k$ |
| Dynamic Routing [21] | Layer33-PSP | - | 270.0 | 79.7 | 180$N$ | 0 | 7.2$k$ |
| ScaleNAS (Ours) | ScaleNet-S1 | **25.3M** | 265.5 | 80.5 | 200 | 400 | 600 |
| ScaleNAS (Ours) | ScaleNet-S2 | 28.5M | 309.5 | **81.1** | **200** | 400 | **600** |
| Auto-DeepLab [23] | Searched-F48-ASPP | - | 695.0 | 80.3 | 72$N$ | 350$N$ | 16.9$k$ |
| HRNet [34] | HRNet-W48 | **65.8M** | 696.2 | 81.1 | - | 260$N$ | - |
| ScaleNAS (Ours) | ScaleNet-S4 | 67.5M | **673.6** | **82.0** | **300** | **600** | **900** |

Table 2: Top-down human pose estimation results.

| Comparison on COCO *val2017*. AutoPose* reports results without ImageNet pretraining. | | | | | | | | | | |
|---|---|---|---|---|---|---|---|---|---|---|
| Method | Backbone | Input size | #Params | GFLOPs | AP | AP$_{50}$ | AP$_{75}$ | AP$_M$ | AP$_L$ | AR |
| SimpleBaseline [35] | ResNet-152 | 256×192 | 68.6M | 15.7 | 72.0 | 89.3 | 79.8 | 68.7 | 78.9 | 77.8 |
| AutoPose [15] | AutoPose* | | - | 10.65 | 73.6 | 90.6 | 80.1 | 69.8 | 79.7 | 78.1 |
| HRNet [34] | HRNet-W48 | | 63.6M | 14.6 | 75.1 | **90.6** | 82.2 | 71.5 | 81.8 | 80.4 |
| ScaleNAS (Ours) | ScaleNet-P2 | | **35.6M** | **8.0** | 75.2 | 90.4 | **82.4** | **71.6** | **81.9** | **80.4** |
| PNFS [36] | PoseNFS-3 | 384×288 | - | 14.8 | 73.0 | - | - | - | - | - |
| SimpleBaseline [35] | ResNet-152 | | 68.6M | 35.6 | 74.3 | 89.6 | 81.1 | 70.5 | 79.7 | 79.7 |
| HRNet [34] | HRNet-W48 | | 63.6M | 32.9 | 76.3 | 90.8 | 82.9 | 72.3 | 83.4 | 81.2 |
| ScaleNAS (Ours) | ScaleNet-P3 | | **26.2M** | **14.3** | 76.3 | 90.7 | 82.9 | 72.5 | 83.3 | 81.3 |
| ScaleNAS (Ours) | ScaleNet-P4 | | 64.3M | 32.6 | **77.0** | **90.9** | **83.6** | **73.0** | **84.2** | **81.8** |
| Comparison on MPII *val*. The GFLOPs is calculated on the input size 256 × 256. We reuse the ScaleNet-P and apply it to MPII . | | | | | | | | | | |
| Method | Backbone | #Params | GFLOPs | mean | Head | Shoulder | Elbow | Wrist | Hip | Knee |
| SimpleBaseline [35] | ResNet-152 | 68.6M | 20.9 | 88.5 | 96.4 | 95.3 | 89.0 | 83.2 | 88.4 | 84.0 |
| HRNet [34] | HRNet-W32 | 28.5M | 9.5 | 90.3 | 97.1 | 95.9 | 90.3 | 86.4 | 89.1 | 87.1 |
| ScaleNAS (Ours) | ScaleNet-P1 | **28.5M** | **9.3** | **91.0** | **97.3** | **96.5** | **91.5** | **87.3** | **90.0** | **87.5** |

**Top-down Methods.** Table 2 summarizes the results of top-down methods on COCO *val2017* and MPII *val*, compared with other state-of-the-art methods. Under 256×192 input resolution, ScaleNet-P2 outperforms manually designed SimpleBaseline [35] (+3.2%) and NAS based AutoPose [15] (+1.6%) by a large margin. In addition, ScaleNet-P2 is comparable with the strong HRNet [34] baseline but with only 56% parameters and 55% FLOPs. With 384×288 input resolution, ScaleNet-P3 achieves 76.3% AP on COCO *val2017*, outperforming PoseNFS-3 [36] by 3.3% AP with less computation cost. ScaleNet-P3 has the same accuracy as HRNet-W48 but uses only 42% parameters and 43% FLOPs. ScaleNet-P4 obtains 77.0% AP, surpassing its strong HRNet counterpart by 0.7% AP. For MPII, ScaleNet-P1 performs the best comparing with SimpleBaseline and HRNet.

**Bottom-up Methods.** As shown in Table 3, by utilizing our ScaleNet-P as feature extractor, we boost the performance of bottom-up pose estimation. Our ScaleNet-P4 and ScaleNet-P1 outperform their counterparts by 0.7% AP and 0.5% AP on COCO *val2017*,respectively. When performing evaluation on COCO *test-dev2017* and CrowdePose *test*, our ScaleNet-P4 achieves a new state-of-the-art. In particular, ScaleNet-P4 achieves 71.3% AP on CrowdPose *test*, surpassing the

Table 3: Bottom-up human pose estimation results.

| Comparison on COCO *val2017* w/o multi-scale test. | | | | | |
|---|---|---|---|---|---|
| Method | Backbone | Input size | #Params | GFLOPs | AP |
| HigherHRNet [10] | HRNet-W32 | 512 | 28.6M | 47.9 | 67.1 |
| | ScaleNet-P1(Ours) | 512 | **28.6M** | **46.9** | 67.8 |
| | HRNet-W48 | 640 | 63.8M | 154.3 | 69.9 |
| | ScaleNet-P4(Ours) | 640 | 64.4M | 141.5 | **70.4** |
| Comparison on COCO *test-dev 2017* w/ multi-scale test. | | | | | |
| Method | Backbone | Input size | #Params | GFLOPs | AP |
| PersonLab [28] | ResNet-152 | 1401 | 68.7M | 405.5 | 68.7 |
| HigherHRNet [10] | HRNet-W48 | 640 | **63.8M** | 154.3 | 70.5 |
| ScaleNAS (ours) | ScaleNet-P4(Ours) | 640 | 64.4M | **141.5** | **71.6** |
| Comparison on CrowdePose *test* w/ multi-scale test. | | | | | |
| Method | Backbone | Input size | #Params | GFLOPs | AP |
| HigherHRNet [10] | HRNet-W48 | 640 | **63.8M** | 154.0 | 67.6 |
| ScaleNAS (ours) | ScaleNet-P4(Ours) | 640 | 64.4M | **141.2** | **71.3** |

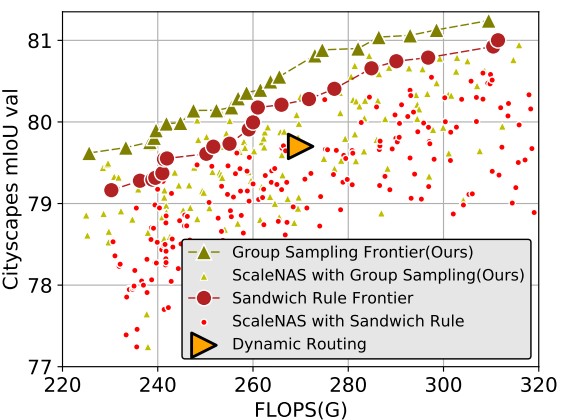

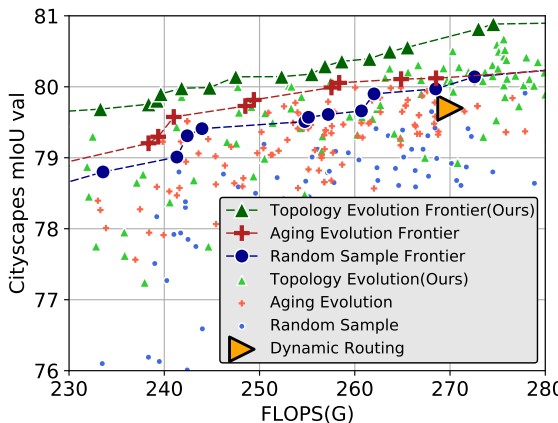

Figure 7: Ablation study of grouped sampling. The Pareto front of grouped sampling steadily higher than the Pareto front achieved by sandwich rule.

Figure 8: Ablation study of topology evolution. The Pareto front of our topology evolution is consistently higher than other existing works.

previous best result by a large 3.7% AP margin. We find that our ScaleNet performs even better with multi-scale test and crowded scenes, suggesting its superior ability to learn scale-aware representation to tackle large scale variance in these scenarios.

### 5.3 Ablation Study

We perform ablation study on each proposed technique. All results are conducted with SuperScaleNet-Seg-W32 on Cityscapes. For simplicity, we denote SuperScaleNet-Seg-W32 as supernet in this part.

**Impact of grouped sampling.** To study the impact of sampling technique, we train two supernets: one is based on our grouped sampling (Section 4.1), the other is based on state-of-the-art sampling method – sandwich rule [38]. We derive the Pareto front from these two supernets based on our proposed evolutionary search. In Figure 7, we show elite architectures and their corresponding accuracy from 220G to 320G. The Pareto front results suggest grouped sampling perform the best for multi-scale aggregation search space which has a wide spectrum of architectures.

**Impact of topology evolution.** We demonstrate the performance contribution from our topology evolution by comparing with random search and aging evolution [30, 6]. We use the same sampling size for each of the method and plot its Pareto front. Aging Evolution is implemented following the hyperparameters in [30]. While aging evolution only has mutation to generate offsprings, our evolution method highly benefits from inner-stage crossover. As shown in Figure 8, under the same searching budget, our multi-scale topology evolution consistently achieves better performance on Pareto front thanks to our inner-stage crossover and mutation techniques.

## 6 Conclusion

We present ScaleNAS, a multi-path one-shot learning method for scale-aware representations. To the best of our knowledge, ScaleNAS is the first of its kind one-shot NAS method that considers scale variance for multiple vision recognition tasks. To efficiently search a wide spectrum of neural architectures for different vision tasks, we rest upon the following key ideas: (i) A novel multi-scale feature aggregation search space that includes cross scale feature fusions and flexible depths. (ii) One-shot based training method driven by an efficient sampling technique to train multi-scale supernet. (iii) Multi-scale architecture topology evolution to efficiently search elite neural architectures. All the above novel ideas coherently make ScaleNAS outperform existing hand crafted and NAS-based methods on semantic segmentation and human pose estimation.

# 7 Limitations and Broader Impact Statement

**Limitation.** We observe that larger elite models require more fusions than blocks. This might impose future design constraints as larger models require more computation cost on feature fusions. The deployability and on-device performance can be further studied as future works.

**Broader Impact.** Designing high resolution representation models is computation consuming which may cause potential environmental impact (e.g., carbon footprint and global warming). This research vastly reduces the design and evaluation time and brings positive environmental impact by enabling a more flexible design space with the proposed efficient one-shot searching method.


## A  Supplementary Material

This supplementary material provides more details of evolutionary algorithm, training on each task and also the extension of object detection task. For *reproducibility*, **we provide full searching and training codes, as well as pretrained models**. Please refer to `README.md` to see detailed instructions.

### A.1  Algorithm Description of Multi-Scale Architecture Topology Evolution

Our *Multi-Scale Architecture Topology Evolution* is described in Algorithm 1. Detail code implementation can be found in `tools/evo_server.py`

---
**Algorithm 1** Multi-Scale Topology Evolution

---
**Input:** Search space $S$, Trained SuperScaleNet, initial population size $n_0$, number of offspring $k$, crossover probability $p_c$, mutation probability $p_m$, number of final elite architectures $N$.
**Output:** $N$ Elite ScaleNet
1: Sample $n_0$ sub-networks to obtain initial population $\mathbf{D} = \{arch_d, d = 1, 2, 3, \ldots, n_0\}$
2: **while** $\text{len}(\mathbf{D}) < N$ **do**
3:     Select top $k$ models on the Pareto front as candidate group $\mathbf{C} = \{arch_c, c = 1, 2, 3, \ldots, k\}$
4:     **for** every sub-networks $\mathbf{arch_c}$ in $\mathbf{C}$ **do**
5:         Do crossover under probability $p_c$
6:         Do mutation under probability $p_m$
7:         Get offspring model $offspring\_arch_c$
8:     Gather $k$ offspring models as set $\mathbf{M_k} = \{offspring\_arch_c, c = 1, 2, \ldots, k\}$
9:     Update $\mathbf{D} = \mathbf{D} \cup \mathbf{M}$

---

### A.2  Details of Search Space Exploration

In our main submission, we conduct initial search space exploration on semantic segmentation using Cityscapes. All models are trained from scratch for 48 epochs. Data augmentation strategies and other training protocols are the same as the teacher training part of Section A.8 in this supplementary material.

We further validate the effectiveness of our search space by studying the influence of *number of feature fusions* and *residual blocks* and compare with HRNet baseline. These models are denoted as ScaleNet-G series as shown in Figure 9. Original HRNet has 108 residual blocks [18] and 62 feature fusions. Residual block is composed of two $3 \times 3$ convolutions. Multi-scale fusion includes downsampling and upsampling. For downsampling, we use strided $3 \times 3$ convolution with stride 2. For upsampling, we use bilinear upsampling followed by a $1 \times 1$ convolution for aligning the number of channels [34].

We create ScaleNet-G1 by using the same number of blocks as HRNet while using 12 less feature fusions with our proposed search space, we observe that there are some ScaleNet-G1 models perform better than HRNet while having less number of fusions. Therefore, the feature fusion position of HRNet may not be optimal. Based on ScaleNet-G1, we create ScaleNet-G2 and ScaleNet-G3

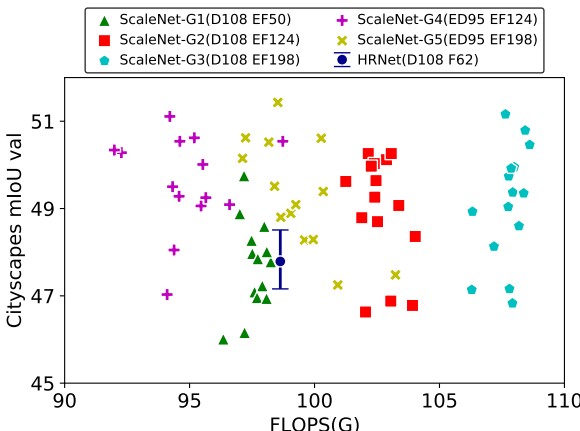

Figure 9: Search space exploration. We train HRNet five times and record the mean and variance of their accuracy. HRNet has 108 resblocks (denoted as 'D') and 62 fusions(denoted as 'F'). We randomly sample 5 groups of ScaleNet with different resblocks and fusions based on our search space. 'ED' and 'EF' represents the expectation of blocks and fusions, respectively.

by increasing the number of fusions to the expectation of 124 and 198, respectively. We notice that more feature fusions comes with a higher mIoU and inevitable comes with higher FLOPs. To study the redundancy of number of blocks, we create G4 and G5 by decreasing number of blocks while keeping higher number of fusions. We observe that the mean accuracy of ScaleNet-G4, G5 are still higher than the original HRNet setting. Based on this observation, we envision that we can use neural architecture search to explore the trade-offs and relationships between blocks and fusion connection.

### A.3 Elite Architecture Pattern Analysis

Different hardware platforms have different computation constraints. We analyze the deployability of elite architectures with different computation cost. We record FLOPs, the number of fusions, and the number of blocks of Pareto front from the 2000 elite ScaleNets collected by our evolutionary method.

In Figure 10(a)(c), we observe larger elite models have more fusions than blocks. In addition, the number of fusions increases faster than the number of blocks. To further analyze the relationship between number of fusions and number of blocks, we demonstrate block-fusion ratio in Figure 10(b)(d).We observe that for the largest elite model, it requires two times more fusions than blocks. However, for small elite models, the number of fusions is only half of the number of blocks. This interesting observation provides important future design insights: 1) For edge devices, we should invest more computation cost on blocks than fusions. 2) To design larger models, it is preferable to invest computation cost on fusions over blocks.

### A.4 Impact of Distillation

We further study whether distillation plays an important role in accuracy gain. Based on the same training procedure in Section A.8, we train ScaleNet-S1 from ImageNet pretrained weights (stand-alone) with and without distillation. We use the mean squared error (MSE) loss with distillation ratio 1 and the teacher model pretrained on Cityscapes. As shown in Table 4, the accuracy of the stand-alone training is only slightly lower than directly taken from supernet. It suggests that distillation is beneficial but not a dominant factor in the final accuracy.

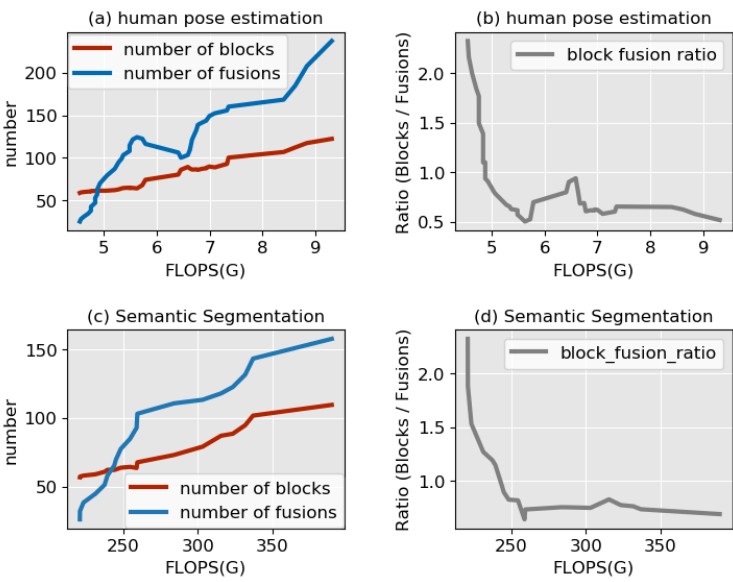

Figure 10: The network pattern of elite sub-networks. We show the relationship between number of blocks and fusions for the elite sub-networks.

Table 4: Ablation study of distillation. Comparison with stand-alone training with and without distillation. The performance mIoU(%) is obtained on Cityscapes *val*.

| Model | from supernet | stand-alone w/o distillation | stand-alone w/ distillation |
|---|---|---|---|
| ScaleNet-S1 | 80.5 | 80.2 | 80.4 |

## A.5 Results on Object Detection

We directly apply the ScaleNet-S2, which is obtained from semantic segmentation, to object detection task. We plug in the ScaleNet-S2 to two classic object detection frameworks, Faster R-CNN [31] and Mask R-CNN [17] as shown in Table 5. We use the whole COCO *trainval135* as training set and validate on COCO *minival*. For both Faster R-CNN and Mask R-CNN, the input images are resized to a short side of 800 pixels and a long side not exceeding 1333 pixels. We use SGD as optimizer with 0.9 momentum. For a fair comparison, all our models are trained for 12 epochs, known as 1× scheduler. We use 8 TESLA V100 GPUs for training with 16 global batch size. The initial learning rate is 0.02 and is divided by 10 at 8 and 11 epochs.

In the Faster R-CNN framework, our networks perform better than HRNet-w32 with less parameters and computation cost. Our ScaleNet-S2 is especially effective for small objects (1.1% improvement for $AP_S$). The reason is that our ScaleNet-S2 learns more high-resolution features which are beneficial for small objects.

## A.6 Details of Found Architectures

Here we provide the architecture structures of our crafted architecture ScaleNet-P1 for human pose estimation and ScaleNet-S1 for semantic segmentation, see Figure 11 and Figure 12, respectively. The interesting observation is that both architectures have more multi-scale feature fusion at later stages while with a relatively simple network structure at the early stages.

## A.7 Details of Training Teacher Model on ImageNet

Following the instructions in [34], we use stochastic gradient descent (SGD) as the optimizer with 0.9 nesterov momentum and 0.0001 weight decay. The model is trained for 100 epochs with batch

Table 5: Object detection results on COCO *minival* in Faster R-CNN [31] and Mask R-CNN [17]. LS denotes learning rate scheduler. GFLOPs is calculated on the input size 800×1280. HRNet-w32* denotes our reimplementation.

| Backbone | Params(M) | GFLOPs | box | | | | mask | | | |
|---|---|---|---|---|---|---|---|---|---|---|
| | | | AP | $AP_S$ | $AP_M$ | $AP_L$ | AP | $AP_S$ | $AP_M$ | $AP_L$ |
| Faster R-CNN [31] | | | | | | | | | | |
| HRNet-w32* | 47.2 | 285.4 | 39.8 | 22.8 | 43.7 | 51.0 | / | / | / | / |
| **ScaleNet-S2** | 46.3 | 271.3 | 40.1 | 23.9 | 44.2 | 51.7 | / | / | / | / |
| Mask R-CNN [17] | | | | | | | | | | |
| HRNet-w32* | 49.9 | 353.9 | 40.8 | 23.8 | 44.5 | 52.4 | 36.4 | 19.5 | 39.7 | 48.9 |
| **ScaleNet-S2** | 49.0 | 339.8 | 40.9 | 24.4 | 44.6 | 52.5 | 36.5 | 19.7 | 40.0 | 49.0 |

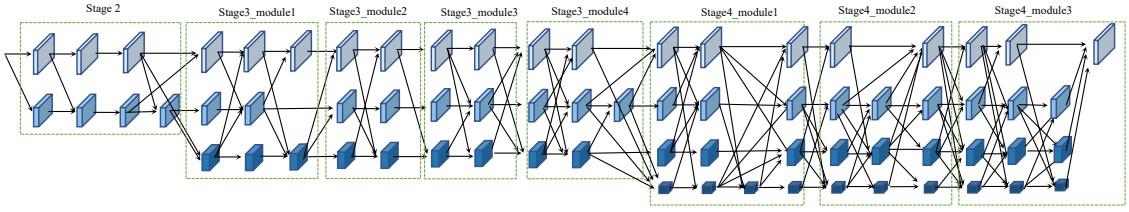

Figure 11: The full model of ScaleNet-S1.

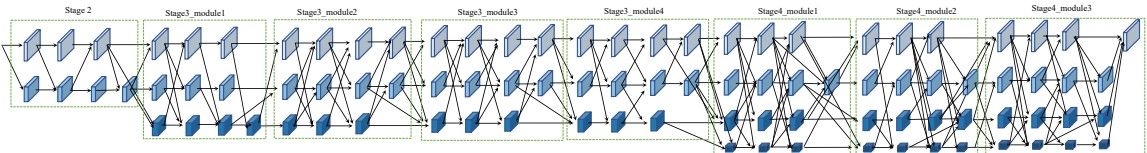

Figure 12: The full model of ScaleNet-P1.

size 768. The initial learning rate is set to 0.3 and is reduced by 10 at epoch 30, 60, and 90. It takes ~30 hours to train on 16 TESLA V100 GPUs.

## A.8 Details of Training SuperScaleNet on Semantic Segmentation

**Cityscapes dataset.** The Cityscapes [13] is a widely used dataset for semantic segmentation tasks, which contains 5,000 high quality pixel-level finely annotated scene images. The dataset is divided into 2975/500/1525 images for training, validation, and testing, respectively. There are 30 classes, and 19 classes among them are used for evaluation. The mean of class-wise intersection over union (*mIoU*) is adopted as our evaluation metric.

**Teacher model training.** For a fair comparison, we follow the same training protocols in [34]. We adopt the SGD optimizer with the momentum of 0.9 and the weight decay of 0.0005. The model is trained for 484 epochs with the batch size of 24 on 8 TESLA V100 GPUs. The initial learning rate is set to 0.01 and the cosine annual decay [25] is used for decaying the learning rate. For data preprocessing, the training and validation image size is 512×1024 and 1024×2048, respectively. For data augmentation strategies, we use random cropping (from 1024×2048 to 512×1024), random scaling (between [0.5, 2]), and random horizontal flipping.

**SuperScaleNet-Seg training:** We follow the same training protocols as teacher training except the initial learning rate is set to 0.001. This is because the SuperScaleNet-Seg is initialized from the well-trained teacher, we only need to fine-tune each sub-network using a small learning rate. It takes ~40 (60) hours to obtain SuperScaleNet-Seg-W32 (W48), including the teacher training. With only

twice the training cost as stand-alone model training, we can obtain a series of segmentation models in a wide spectrum of FLOPs without additional retraining. We further use multi-scale topology evolution to explore elite ScaleNet-Seg.

### A.9  Details of Training SuperScaleNet on Top-Down Human Pose Estimation

**COCO dataset.** We train SuperScaleNet-Pose on COCO [22] *train2017* dataset (57K images and 150K person instances) and evaluate it on COCO *val2017*. To evaluate object keypoints, we use Object Keypoint Similairty (OKS). We break down the performance on different OKS: $AP_{50}$ and $AP_{75}$. We also report the performance on different sizes of object. $AP_M$ and $AP_L$ stands for AP of medium object and large object, respectively.

**MPII dataset.** The MPII Human Pose dataset [1] consists real-world full-body pose and annotations. There are around 25K images with 40K subjects, where 12K subjects are used for testing and the remaining subjects are used for training. We use the PCKh (head-normalized probability of correct keypoint) score as our evaluation metric, following [35, 33]

**Teacher model training.** Following the training protocols of HRNet [34], we train the model for 210 epochs using the Adam optimizer [19] with step learning rate decay [35, 34]. The initial learning rate is set as 0.001, and is dropped to 0.0001 and 0.00001 at the 170th and 200th epochs, respectively. For data preprocessing, we extend the human detection box in height or width to a fixed aspect ratio – height : width = 4 : 3, and then crop the box from the image, which is resized to a fixed size, $256 \times 192$ or $384 \times 288$. For data augmentation strategies, we use random rotation ([-45°, 45°]), random scale ([0.65, 1.35]), and flipping.

**SuperScaleNet-Pose training.** We follow the same training protocols in teacher training. We do not reduce the learning rate as in SuperScaleNet-Seg training because the Adam optimizer can adjust the learning rate adaptively [19]. The models are trained on 8 TESLA V100 GPUs. It takes ~50(75) hours to train SuperScaleNet-Pose-W32(W48), including the teacher training. After topology evolution, we further fine-tune the ScaleNet-P for 20 epochs (around 3 hours) for better performance.

**MPII training.** We use the same data augmentation and training strategy for MPII, except that the input size is cropped to $256 \times 256$ for a fair comparison with SimpleBaseline [35] and HRNet [33].

### A.10  Details of Training Found Architectures on Bottom-Up Human Pose Estimation

**CrowdPose dataset.** The CrowdPose [20] dataset consists of 20K images, containing about 80K persons. The training, validation and testing subset are split in proportional to 5:1:4. CrowdPose has more crowded scenes than the COCO keypoint dataset, posing more challenges to pose estimation methods. The evaluation metric is the same as COCO [22].

**Implementation details.** We adopt the standard training procedure on COCO *train2017* and CrowdePose *trainval* as in [26, 10], then report results on COCO *val2017/ test-dev2017* and CrowdePose *test*. The models are trained for 300 epochs. We train ScaleNet-P series on bottom-up human pose estimation framework, HigherHRNet [10]. For a fair comparison, we use the exact same training routine as HigherHRNet. Specifically, we train our model for 300 epochs using the Adam optimizer [19] with step learning rate decay. The base learning rate is set to 0.001, and dropped to 1e-4 and 1e-5 at the 200th and 260th epochs, respectively. For data augmentation strategies, we use random rotation ([-30°, 30°]), random scale ([0.75, 1.5]), random translation ([-40, 40]), random crop ($512 \times 512$), and random flip. We use the top-down SuperScaleNet-Pose to initialize weights.

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
