# OpenReview forum: "ScaleNAS: Multi-Path One-Shot NAS for Scale-Aware High-Resolution Representation"
_automl.cc/AutoML/2022/Track/Main — AutoML-Conf 2022 (Main Track)_

### Official Review · Reviewer_dwPB · 2022-03-23

**Potential Impact On The Field Of Automl Rating:** 3
**Technical Quality And Correctness:** I see no concerns on technical correc…
**Technical Quality And Correctness Rating:** 4
**Clarity:** The paper is clear, easy to follow, a…
**Clarity Rating:** 3

**Summary Of Contributions:**

The paper describes a NAS solution for dense prediction tasks, such as semantic segmentation and human pose estimation. The primary intellectual difference is the search space, where the authors considered a multi-stage, cross-scale, generic search space. Experiments are performed on benchmark datasets in semantic segmentation and human pose estimation.

**Overall Review:**

I do believe that NAS-for-dense-prediction is relatively overlooked compared against NAS-for-image-classification. And the best way to do multi-scale is the primary difference. This paper considers a new search space that is more general and challenging than before. Though the proposed approach has flavors from many existing works such as OFA, BigNAS etc, I think the overall it is a good contribution to the AutoML community. Ablation studies are performed to show the impact of both proposed components (grouped sampling and topology evolution), and the end result performs competitively, especially on human pose estimation.

**Potential Impact On The Field Of Automl:**

The paper proposes a generic search space for dense image prediction problems, and sets the state-of-the-art on some problems.

**Reproducibility:**

It should be possible to reproduce the work based on the paper, though I did not see the plan or claim to release code. Also, the architecture of the found ScaleNAS nets are not illustrated or discussed, which is a big issue and concern.

**Review Confidence:**

5: You are absolutely certain about your assessment. You are very familiar with the related work and checked all the details carefully.

**Review Rating:**

5: Accept, good paper

**Review Summary:**

I recommend acceptance, though I strongly suggest the authors add illustration and discussion of the found ScaleNAS nets for reproducibility and insights.

---

### Official Review · Reviewer_n6cc · 2022-04-01

**Potential Impact On The Field Of Automl Rating:** 3
**Technical Quality And Correctness:** The approach, the experiments, and co…
**Technical Quality And Correctness Rating:** 4
**Clarity Rating:** 3

**Summary Of Contributions:**

The authors propose a one-shot NAS method to learn multi-branch architectures with high-resolution representations that can be applied to different visual recognition tasks. To this aim they design a search space based on HRNet with learnable depth and branching pattern. The search is done in two-stages consisting of supernet training and architecture search. For supernet training, they propose grouped sampling to enhance exploration during the sampling of architectures from the supernet. For the architecture search, a multi-scale topology evolution is proposed with a crossover step that acts between stage modules of the network. Due to the two-stage procedure, this also allows to extract all pareto optimal architectures instead of a single architecture as in previously proposed methods.

**Clarity:**

The paper is presented in a clear way in most parts. Two parts that could benefit from more clarity are

•	Formula 1 and line 194: While it is clear from the context, the abbreviation “MSE” could be spelled out once

•	Grouped sampling: The example regarding the depth choice is in my opinion not clear. In particular, it seems to mix different notations (i.e., what is described with curly braces in lines 210-211).

•	Crossovers in multi-scale evolution (line 232ff): The definition of the stage module is a bit hidden in the text and could maybe be illustrated in figure 4 as well,

**Overall Review:**

Strengths:

1. Search space: The design of the search space around HRNet allows it to be applied to many visual recognition tasks, in particular also those that require high-resolution features. In addition, the search space has a very flexible macro-architecture with learnable branching pattern.

2. Experimental results show good performance improvements over other NAS algorithms and baselines on several datasets while still maintaining a relatively low search cost which is further amortized in the case of many deployment scenarios.

Weaknesses:

1. The search space is very flexible regarding the macro-architecture (i.e., the structure within the different stages) but in its current form does not consider the micro architecture (e.g., operations or width of hidden representations). In addition, scale variance and high-resolution representation learning based on HRNet has e.g., been explored in HR-NAS (https://openaccess.thecvf.com/content/CVPR2021/html/Ding_HR-NAS_Searching_Efficient_High-Resolution_Neural_Architectures_With_Lightweight_Transformers_CVPR_2021_paper.html) which has a less flexible branching pattern in each stage but can e.g., learn the number of feature maps in the different resolution branches. This work is unfortunately not mentioned at all and a detailed comparison with advantages and disadvantages of ScaleNAS compared to HR-NAS would be important. In addition, the main strength, namely the high flexibility in the macro-architecture is not really discussed in the experiments in the main paper. How many branches are e.g., used in the top-performing architectures from the experiments?

2. Grouped sampling and the sandwich rule: Grouped sampling is phrased as an alternative to the sandwich rule. However, isn’t it rather an alternative to the random sampling of an architecture which is done as part of the sandwich rule? I.e., couldn’t the largest and smallest architecture still be sampled in addition to the architecture sampled with grouped sampling? Is there something like the “smallest” architecture in the case of the proposed search space?

3. Grouped sampling procedure: Grouped sampling is motivated by stating that the architecture distribution is approximately normal according to the central limit theorem. However, the architecture distribution is a complex compound distribution, and it is not directly clear to me why the central limit theorem applies. E.g., for SPOS, this is not the case as all architectures are equally likely. The description of the grouped sampling for branch depth is also not quite clear to me (see also “Clarity” section). Does the grouped sampling cover the whole search space?

4. Relation to MixPath: One results of MixPath (https://arxiv.org/abs/2001.05887) is that if you sum the feature vectors from multiple path those are close to multiples of the feature vectors from different paths which is causing training instability. Is this an issue in your search space as well as the number of inputs to a feature map is not fixed and why? If not, wouldn’t it be possible to directly compare to MixPath as you already have the trained teacher network where you could add the architecture parameters?

5. Discussion and comparison of some relevant related work seems to be missing, e.g., a.)	HR-NAS (see point 1) b.)	The grouped sampling strategy that divides the search space into smaller sub-search seems to be similar to Few-shot NAS (https://arxiv.org/abs/2006.06863) and there are also other approaches such as AttentiveNAS (https://openaccess.thecvf.com/content/CVPR2021/papers/Wang_AttentiveNAS_Improving_Neural_Architecture_Search_via_Attentive_Sampling_CVPR_2021_paper.pdf) that address the sampling in the sandwich rule. Again, the paper would benefit from these a more detailed discussion and comparison with these approaches. c.)	FasterSeg (https://openreview.net/forum?id=BJgqQ6NYvB) could be added as an additional multi-branch method

Minor:

1. Reading the formulation “ScaleNAS solves multiple tasks at a time” (line 10) could be mistaken to think that the method is about multi-task networks and not of the fact the method can be applied to multiple tasks

2. Are 40 deployment scenarios (line 259) realistic? Even for much smaller numbers the proposed approach already needs less training time.

Spelling:

•	line 239: “accuraices” -> accuracies

•	line 259: “s” -> is

**Potential Impact On The Field Of Automl:**

The paper addresses a the highly relevant problem of high-resolution representation learning using a search space based on a state-of-the-art computer vision model.

**Reproducibility:**

The reproducibility of the results seems to be possible as reproducibility checklist is filled out and the full training code as well as pretrained models are provided.

**Review Confidence:**

3: You are fairly confident in your assessment. It is possible that you did not understand some parts of the submission or that you are unfamiliar with some pieces of related work.

**Review Rating:**

5: Accept, good paper

**Review Summary:**

The authors tackle the problem of applying one-shot NAS to learn high-resolution representations for different visual recognition tasks and demonstrate improved performance on several datasets compared to other NAS methods. The main weakness in my opinion is the missing comparison (or at least discussion) to the very similar HR-NAS paper, the limited micro architecture of the search space and the clarity of the grouped sampling approach. This leads me to reject the paper in its current form.

---

### Official Review · Reviewer_HYZy · 2022-04-03

**Potential Impact On The Field Of Automl Rating:** 3
**Technical Quality And Correctness Rating:** 4
**Clarity Rating:** 3

**Summary Of Contributions:**

The paper presents ScaleNAS, a Neural Architecture Search method, to find an optimized architecture for visual recognition tasks.
The search space of the method is *multiscale*: each branch can have different depths, as opposed to searching the overall depth for the entire network, and each feature maps can be randomly connected to each of the other scales of feature maps.
To sample a novel architecture, the authors propose a *grouped sampling* method. The search space is divided in several subspaces (here 9) to cover the full space, and sampling is done from each of the subspace.
At the beginning of training, the model and a teacher model are initialized by the largest architecture from the search space pretrained on ImageNet.
At each iteration of training, a novel architecture is sampled following the grouped sampling approach, and trained using the true labels of the given task and soft labels from the teacher network. The full trained model is called *SuperScaleNet*.
After training, the best subnetworks from *SuperScaleNet* are searched using an evolutionary method, that allows for crossover of modules between subnetworks and mutation of connections.
The experiments are done on the Semantic Segmentation and Human Pose Estimation tasks, where subnetworks found by ScaleNAS outperforms state-of-the-art methods on each task.

**Clarity:**

The paper is dense with a lot of information. The figures help better understand the workflow of the method, but I think the full training procedure should be more detailed. The differences between the ScaleNet, SuperScaleNet and Elite ScaleNet should be more explicitly explained. For the Group Sampling strategy, how the sampling from each subgroup is done and which subgroup to sample from is not really described. I had to specifically look in the code to understand.

**Overall Review:**

### Positive

- The training of a single super network (one-shot training) and sampling subnetwork greatly reduces the total computation cost of the method.
- The architectures found outperforms state-of-the-art methods on each task (semantic segmentation and human pose estimation), with lower number of parameters and GFLOPs.
- Both Group Sampling and Topology Evolution are more effective searching strategies and improve over other used strategies.

### Negative

- The method is not really *multitask*, as it trains a super network and find subnetworks tailored for a given task. Even though the search space and the method is the same for each task, the architectures found are not the same between tasks.

**Potential Impact On The Field Of Automl:**

The huge computation cost of Neural Architecture Search is a major downside and currently limits its widespread use. However, the One-Shot training approach of this method greatly reduces the total computation cost, which is important for its democratization.

**Reproducibility:**

All the hyperparameters are detailed in the supplementary material, and the code was also made available by the authors.

**Review Confidence:**

3: You are fairly confident in your assessment. It is possible that you did not understand some parts of the submission or that you are unfamiliar with some pieces of related work.

**Review Rating:**

5: Accept, good paper

**Review Summary:**

The method presented in the paper incorporates several techniques that improve the searching and sampling strategies for Neural Architecture Search. The paper also show that one-shot training for multi-path and high resolution tasks can achieve state-of-the-art results for a lower total computation cost.

**Technical Quality And Correctness:**

There is a lot of technical novelty in the paper. The ablative studies show the impact of each proposed techniques, the group sampling strategy and the topology evolution approach.

---

### Official Review · Reviewer_RyMt · 2022-04-04

**Potential Impact On The Field Of Automl Rating:** 4
**Technical Quality And Correctness:** N/A for reproducibility reviewers
**Technical Quality And Correctness Rating:** 4
**Clarity:** N/A for reproducibility reviewers
**Clarity Rating:** 3

**Summary Of Contributions:**

The main contribution of ScaleNAS is that it presents a novel way of utilising multiple representations of different scales by employing grouped sampling and evolutionary search. Doing so, they significantly boost the robustness of the segmentation/pose estimation models they discover since they make the representations scale-invariant, which is crucial for real-world applications. The authors also design a flexible search space to explore different depths of network layers for different scales.

Arguably, the most important contribution of the paper is its novel multi-path one-shot search method. They address the caveats of current one-shot methods (namely the limitation for training on a single path) by proposing a multi-scale aggregation search space, which in turn unveils new challenges like the difficulty of exploring the new colossal search space. To overcome this they use elite architectures, evolutionary approaches and group sampling.

**Overall Review:**

The paper introduces a novel way to substantially reduce the needed computational power needed for NAS and also presents an application of NAS to pose estimation, which is also a niche area. It combines multiple state-of-the-art methodologies to achieve what seems to be a similar way to Miao, Caijing, et al.'s "Sixray: A large-scale security inspection x-ray benchmark for prohibited item discovery in overlapping images.", but instead of helping a model just to have different auxiliary scales for the sake of addressing item crowding, they introduce a flexible search space containing a multitude of models and manage to beat the state-of-the-art in two separate common datasets with a strong performance increase.

I believe the findings of the paper are of tremendous importance for the field, yet I would like to point out that using one-shot learning with SuperNets can lead to networks of immense size, which makes training such networks not only harder and slower but also hinders the general interpretability of networks. The approach can be benefitted from the introduction of a pruning method to discard paths that are not promising, which can also help with the reduction of the search space, which even though reduced is quite big (which is normal for any NAS approach).

**Potential Impact On The Field Of Automl:**

The impact of this paper is tremendous since it presents a novel way of using multi-path one-shot search not only in computer vision in general but also in high demand segmentation as well as pose-estimation problems. The field of computer vision has been moving towards the use of multi-scale representations and the ability to boost the interpretability of neural networks through well-defined feature vectors. The paper utilises feature vectors implicitly by using scale-aware representations. This work shows how current state-of-the-art can be improved significantly without rethinking the convolutional networks from scratch. This is important since this drives networks to be more equivariant than invariant to transformations, which is one of the key advantages of capsule networks. I would be interested to see if there a modification can be made such that automatic multi-scale capsules can be discovered.

**Reproducibility:**

The completed reproducability checklist has been filled. An explanation for each Yes or No answer was required by the organisers, which has not been done, but I believe the authors thought they should only provide explanations when they give a negative answer in the checklist.

The conducted experiments and the provided code follow well-established conventions with a detailed and well-written README, which clearly explains how to reproduce the results. All requirements, environments and datasets are defined and provided. Links as well as instructions as to how to get external resources are provided.

The experiments are clearly defined and a comparison of multiple state-of-the-art approaches is provided for a fair comparison. The authors even provide a trade-off graph depicting the trade-off of computational resources with performance across the training cycle of the selected approaches, which provides readers with insights as to what is the improvement in performance over time and what computational resources used they can expect.

I would have loved to see the results of ScaleNAS on NASBench or similar benchmarks, but I'm aware there is non that can facilitate what the authors showcase. A possible way to have a surrogate loss is through using a proxy score such as Synaptic flow score or SNIP, which are usually used in the context of network pruning but have recently been used as proxy performance estimation tools for how good a certain NN is.

Pseudocode for the evolutionary component is also provided even though it is not too elaborate and authors can add a more elaborate explanation as to what parts of the scripts are responsible for the different tasks. One constructive comment would be to document the functions used in the scripts more clearly.

The only thing missing are tests such as unit or integration tests of the whole package, but given that the package is simply provided for reproducibility evaluation and it is not fully released, I would imagine authors will possibly provide these upon acceptance. This is not a requirement from the conference and it would be unfair to deduct anything based on this. I simply share a possible way of improving even further the impeccable work the authors have done.

Also an additional figure and small discussion can be added addressing the final ScaleNAS architecture and what insights were learnt from it.

**Review Confidence:**

3: You are fairly confident in your assessment. It is possible that you did not understand some parts of the submission or that you are unfamiliar with some pieces of related work.

**Review Rating:**

6: Strong accept, should be highlighted

**Review Summary:**

A novel method of incorporating clustered sampling and evolutionary computation with scale-aware representation search is presented. The approach is well explained with numerous different experiments proving its efficiency and also ablation studies for some of the components.

The code is of good quality and although some comments/tests and other qualitative components are missing or neglected, they are compensated with a well structured, clear and elaborate README file, which makes it easy for practitioners with the right hardware to rerun and even build upon the results of the authors.

Overall the paper is written of a professional standard and reads like a Google or Facebook research paper.

---

### Official Review · Reviewer_Cxop · 2022-04-11

**Potential Impact On The Field Of Automl Rating:** 3
**Technical Quality And Correctness:** The claims are sufficiently supported…
**Technical Quality And Correctness Rating:** 4
**Clarity Rating:** 4

**Summary Of Contributions:**

The paper introduces a new one-shot neural architecture search method that addresses the issue of scale variance in computer vision problems by allowing more flexibility in the multi-scale aggregation and network depth. To ensure an efficient exploration, the authors propose grouped sampling to help the supernet training and multi-scale topology evolution to help the architecture selection. The method is evaluated on semantic segmentation and pose estimation and the authors report  improvement over existing related methods.

**Clarity:**

Apart from small typos that can be easily fixed, the paper is written in a clear way.

**Overall Review:**

strengths:
The paper introduces a new method that allows using one-shot supernet for typically resource intensive computer vision tasks like semantic segmentation. It shows the importance of multi-scale fusion not only at the last layer but also for the intermediate layers and extends the auto-deeplab design to support this property  as well as flexible bloc depth. Furthermore, it enable multi-path search for better space exploration.

weaknesses:
Minor formatting issues and typos (missing bold font for table 3, Cityscape on line 250)

**Potential Impact On The Field Of Automl:**

The paper presents an appealing method for the computer vision practitioners since it provides an explicit and systematic way for optimizing  the depth for resolution blocks and raises attention for the multi-scale aggregation at intermediate features level.
Using the group sampling as a substitute for the sandwich rule for training the supernet points out the issue of space exploration under the sandwich rule and it is inspiring for further analysis and potentially opens a door for further optimization.

**Reproducibility:**

The presented experiments and results are fairly reproducible with minor effort.

**Review Confidence:**

4: You are confident in your assessment, but not absolutely certain. It is unlikely, but not impossible, that you did not understand some parts of the submission or that you are unfamiliar with some pieces of related work.

**Review Rating:**

5: Accept, good paper

**Review Summary:**

This paper focuses on an interesting topic and proposes two  components that may facilitate the research area. The proposed methods are reasonable and well evaluated. I think this work is a good submission and should be accepted.

---

### Meta-Review · Area_Chair_YaoP · 2022-05-07

**Recommendation:** Accept
**Confidence:** 4

**Metareview:**

This work focuses on neural architecture search (NAS) for dense prediction. The proposed method, called ScaleNAS, considers a multi-stage, cross-scale search space and consists of two phases: supernet training with grouped sampling and multi-scale topology evolution. Results provided for semantic segmentation and human pose estimation show improved performance relative to previous NAS methods.

Reviewers highlighted the significance of NAS for high-resolution representation learning and how the proposed ScaleNAS considers a general and more challenging search space than previous work. Experimental quality is another strength as ScaleNAS is shown to improve performance at a low search cost. Ablations effectively demonstrate the impact of the proposed grouped sampling and topology evolution procedures. Reviewers agreed that the work is generally reproducible.

In terms of weaknesses, the description of the training procedure could be improved.

Overall, this paper is of high quality and likely to be of interest and significance to AutoML community. Therefore I recommend acceptance.

---

### Decision · Program_Chairs · 2022-05-13

Accept